# Modulation of the Endomembrane System by the Anticancer Natural Product Superstolide/ZJ-101

**DOI:** 10.3390/ijms24119575

**Published:** 2023-05-31

**Authors:** Phillip R. Sanchez, Sarah A. Head, Shan Qian, Haibo Qiu, Avishek Roy, Zhendong Jin, Wei Zheng, Jun O. Liu

**Affiliations:** 1National Center for Advancing Translational Sciences (NCATS), National Institutes of Health, Rockville, MD 20892, USA; phillip.sanchez@nih.gov; 2Department of Pharmacology & Molecular Sciences, Johns Hopkins School of Medicine, Baltimore, MD 21205, USA; 3Department of Pharmaceutical Sciences and Experimental Therapeutics, College of Pharmacy, The University of Iowa, Iowa City, IA 52242, USAzhendong-jin@uiowa.edu (Z.J.)

**Keywords:** natural product, superstolide, glycomics, transcriptomics, phenotypic analysis, 3D spheroid

## Abstract

Marine natural products represent a unique source for clinically relevant drugs due to their vast molecular and mechanistic diversity. ZJ-101 is a structurally simplified analog of the marine natural product superstolide A, isolated from the New Caledonian sea sponge *Neosiphonia Superstes*. The mechanistic activity of the superstolides has until recently remained a mystery. Here, we have identified potent antiproliferative and antiadhesive effects of ZJ-101 on cancer cell lines. Furthermore, through dose–response transcriptomics, we found unique dysregulation of the endomembrane system by ZJ-101 including a selective inhibition of O-glycosylation via lectin and glycomics analysis. We applied this mechanism to a triple-negative breast cancer spheroid model and identified a potential for the reversal of 3D-induced chemoresistance, suggesting a potential for ZJ-101 as a synergistic therapeutic agent.

## 1. Introduction

*Neosiphonia Superstes* is a species of marine sponge localized to the region of New Caledonia in the South Pacific [1]. Originally identified during the 1874 voyage of the HMS Challenger expedition, the sponge was resurrected in the early 1990s when D’auria et al. isolated novel macrolides they termed superstolides A and B, which were subsequently shown to display anticancer activity [2,3,4]. The unprecedented chemical structures of this group of marine natural products suggest they might have a unique cellular target(s) and a novel mechanism of action. However, the scarcity of natural products seriously hampered the biological investigation. In 2013, a truncated superstolide A was designed and synthesized (named ZJ-101) that maintains the potent anticancer activity of the original natural product, thereby solving the supply problem of superstolide A, albeit indirectly [5]. Subsequently, several analogs were produced with enhanced potency in the single-digit nanomolar range. However, the mechanism of action of ZJ-101 and other analogs has remained unknown.

Employing several phenotypic assays in cancer cell lines, we identified novel activities and phenotypes associated with ZJ-101 that have revealed a unique mechanism of action. Notably, an anti-adhesive effect was identified, which was approximately 30-fold more potent than the previously described anti-proliferative effect. Cell cycle and growth rate analysis revealed a strong cytostatic effect. Transcriptomic and lectin stain imaging analyses further converged on a dysregulation of glycosylation within the endomembrane system. Through this analysis, we identified a selectivity for dysregulation of *O*-linked glycosylation, which primarily occurs within the Golgi apparatus. The disruption of glycosylation by ZJ-101 leads to pleiotropic effects on proliferation and adhesion, which synergize with etoposide in a 3D spheroid combination model. Taken together, these findings implicate a unique mechanism of action for the family of superstolides and other related analogs [3,4,5,6].

## 2. Results

### 2.1. ZJ-101 Is Cytostatic

The anti-adhesive effects of ZJ-101 (chemical structure shown in Figure 1A) were first observed using light microscopy. One micromole ZJ-101 overnight treatment of HEK293T cells causes a dramatic effect on morphology, with cells rounding and dislodging from the culture surface (Figure 1B). Although this effect is observed in every cell line tested at similar time points, HEK293T cells provide the clearest example of this morphological change due to their naturally spread appearance in 2D culture (Figure 1B). Treated MDA-MB-231 cells exhibiting the same morphological changes were harvested, and cell cycle analysis via propidium iodide staining was performed. No statistically meaningful differences between negative control and ZJ-101 treated cells were observed either at 24 or 48 h of sustained treatment; taxol, utilized as a positive control, caused a G2-phase arrest as expected (Figure 1C).

To determine the extent to which ZJ-101 induced cell death, we used the growth rate normalized inhibition metrics assay (GR assay) [7]. The GR assay counts populations of live and dead cells using separate live-cell imaging dyes. We found that ZJ-101 acted in a purely cytostatic manner, consistent with a GR value greater than 0, with ZJ-101 achieving a GR of 0.5 at the highest dose after a 72 h treatment in MDA-MB-231 cells (Figure 1D). The EC_50_ of 29–96 nM as measured by the GR assay is nearly identical to that of traditional viability dyes such as resazurin (Appendix A). Two known cytotoxic compounds, triptolide and flavopiridol, were utilized as controls and found to pass below the GR value of 0, indicating cytotoxicity. Additionally, the GR assay was used to evaluate the effect of ZJ-101 on isogenic p53 and p21 knockout HCT-116 cells (Appendix A). p53^−/−^ HCT-116 shared nearly identical GR and EC_50_ values for ZJ-101 with wild-type HCT-116. However, double knockout cells for p21 potentiated the anti-proliferative effect of ZJ-101, resulting in a lower GR value of 0.08 at the highest dose. Despite the observed enhanced suppression of cell growth, the EC_50_ for p21^−/−^ cells is between 115 and 197 nM, compared to the WT values of 29–96 nM.

### 2.2. ZJ-101 Is a Potent Inhibitor of Cell–Cell Adhesion

To discern the effects of cell–surface vs. cell–cell adhesion, we performed a three-day 3D spheroid formation assay. We selected MDA-MB-231 cells for this assay due to their widely reported ability to form tight 3D spheroids in culture [8]. In this assay, cells are seeded at a density of 2 × 10^3^ cells/well in 96-well ultra-low attachment (ULA) plates.

Cells will transition from monolayers at the bottom of each well into 3D spheroids over a period of 72 h, as noted in the negative control (Figure 2A). To measure the compactness of cells into spheroids, termed spheroidicity, we utilize the overall diameter of cellular distribution within the well. Concurrent treatment with ZJ-101 following cell seeding had a dose-dependent effect on spheroid formation at concentrations of up to 5 μM (Figure 2B). Measuring the overall diameter of the spheroid, relative to a negative control, we determined the EC_50_ for spheroid formation inhibition was between 0.69 and 1.16 nM (Figure 2B). We further assessed the ability of ZJ-101 to affect cell–cell contacts already established in 3D spheroids. Using pre-formed spheroids, grown for 72 h prior to drug addition, we observed an equally strong disruption of cell–cell adhesion. At concentrations above 20 nM, ZJ-101 was able to fully dissolve spheroids over 72 h, leaving individual cells to settle at the bottom of the well (Figure 2C). A rapid disassembly of spheroids can be seen even after the first 24 h of treatment (Figure 2C). Quantitation of spheroidicity at 72 h after addition results in an EC_50_ for spheroid disassembly between 2.37 and 12.05 nM. The discrepancy between approximate EC_50′_s for spheroid formation versus disassembly is 4 nM. Both effects, however, remain an order of magnitude smaller than the cytostatic effect observed in the GR assay (Figure 1D).

### 2.3. Transcriptome Analysis Identifies a Dysregulation of the Endomembrane System

Transcriptome analysis has recently become a powerful method to discern drug mechanisms of action. We performed two sets of analyses with four logs of ZJ-101 dose (0.5, 5.0, 50.0, and 500 nM) each. This dose range provides the broadest context for the previously observed phenotypes. To identify differentially expressed genes before cellular morphology is typically affected by ZJ-101, we used a 6 h time point for 2D-cultured MDA-MB-231 cells. Additionally, we performed a 24 h time treatment in pre-formed 3D MDA-MB-231 spheroids to identify genes with altered expression following cell–cell adhesion loss. As previously observed in the spheroid disassembly assay (Figure 2C), this 24 h treatment results in moderate adhesion loss, which increases with dose. Both sets of transcriptome data shared dose-dependent effects on genes within the endomembrane system (GO:0012505), endoplasmic reticulum (GO:0005783), and cell adhesion (GO:0007155), as determined by GO term enrichment analysis via g:Profiler (Figure 3A,B) [9].

The most significantly upregulated genes shared between both sets of data were ATP6V0A1, GPRC5A, SLC3A2, EPHA2, and HSP90AA1 (Figure 3C,D) [10]. Notably, ATP6V0A1, which encodes a V-type ATPase involved in endosome pH regulation, is significantly upregulated by ZJ-101 treatment. Cellular chaperones such as HSP90AA1 and various Hsp40 family members like DNAJB9 are likewise upregulated in both cellular contexts (Figure 3C,D). At 500 nM, the highest concentration assessed, an upregulation of heat-shock responsive genes was observed within the 6 h time point for the 2D cultured cells (Appendix A). Unfolded protein response (UPR)-related target genes are simultaneously suppressed (Appendix A).

Broadly, genes involved in cell adhesion and extracellular matrix organization are downregulated by ZJ-101 treatment (Figure 3C,D). These include the integrin ITGA5, integrin ligand ICAM1, Von Willibrand factor A (VWA1), and collagen type V alpha 1 (COL5A1). Genes involved in protein glycosylation found in the Golgi apparatus, such as B4GALT1 and GALNT1, were also suppressed (Figure 3C,D).

### 2.4. Golgi Function Is Dysregulated by ZJ-101 

Following transcriptome analysis, our focus narrowed to the endomembrane system, consisting of the ER, Golgi, and general secretory vesicles. We first performed immunostaining of the *cis*-Golgi marker GM130 in fixed HeLa cells to observe changes in Golgi morphology at short time points. In HeLa cells, GM130 assumes a compact perinuclear position, providing a simple marker for alterations in morphology. To minimize changes to overall cellular morphology from the cytotoxic positive control, Brefeldin A (BFA), a four-hour time point was chosen. As seen in Figure 4A, four hours of treatment with BFA drastically alters Golgi morphology by redistributing GM130 to the endoplasmic reticulum, while ZJ-101 at a dose of 1 μM does not (Appendix A). Washout of BFA after pre-treatment with ZJ-101 also does not alter the ability of GM130 to recover its original morphology (Appendix A). To interrogate Golgi function, we utilized the fluorescent conjugate of the lectin *Helix Pomatia* Agglutinin (HPA), which selectively binds O-GalNAc residues localized to the Golgi [11,12]. HPA lectin staining was dose-dependently suppressed by ZJ-101 following four hours of treatment in HeLa cells, with the EC_50_ for this effect being approximately 67 nM (Figure 4B). Other lectins, concanavalin A (ConA) and wheat germ agglutinin (WGA), which bind selectively to N-glycans, were unaffected by the treatment (Figure 4B).

Glycomics analysis was performed on both *N*- and *O*-glycans purified from cells treated with 500 nM ZJ-101 for 6 h. Generally, both *N*- and *O*-glycans decreased in relative abundance upon treatment with ZJ-101 (Figure 4C). Several high-molecular-weight *N*-glycans, containing complex polysialylated LacNAc chains, were increased (Table 1). Notable *O*-glycans that were increased include Core-2 *O*-glycans and Disalyl T-Antigen (Table 1).

Glycan incorporation was assessed using an image-based bioorthogonal click-chemistry assay. Azido-modified GlcNAc, GalNAc, and ManNAc, as well as alkynyl-modified Fucose, were added to the media during the 6 h treatment with increasing doses of ZJ-101. Significant decreases in GalNAz and fucose alkyne were observed at 500 nM treatment, with reductions of 12.8% and 25.5% for each, respectively (Figure 4D). No significant changes were observed for GlcNAz incorporation. ManNAz incorporation increased marginally by approximately 12% (Figure 4D).

### 2.5. ZJ-101 Modulates the Endolysosome

We found no disruption of Golgi structure by ZJ-101 treatment, as judged by GM130 staining (Figure 4A), and no dysregulation of endosome pH using lysotracker dye (Figure 5). We utilized multiple controls for the lysotracker assay to compare ZJ-101 with known modulators of the endomembrane system. Neither Kifunensine, an inhibitor of *N*-glycosylation, nor Brefeldin A, a disruptor of Golgi structure and function, decreased lysosome spot intensity relative to the negative control (Figure 5B) [13,14]. Bafilomycin A1, an inhibitor of V-ATPases found on endosomes, potently decreases lysotracker spot intensity, indicating increased endosome pH (Figure 5B) [15]. Although ZJ-101 does not affect lysosome staining intensity, the average lysosome puncta area is significantly (*p* = 0.02) decreased (Figure 5C) and is shared by brefeldin A and kifunensine treatments. Despite the decreased puncta size, the percentage of total area that lysosomes occupy in the cell was not changed by ZJ-101, indicating possible enhanced endolysosomal fission events (Figure 5D).

### 2.6. ZJ-101 Reverses Etoposide Resistance in 3D Spheroid Model

Resistance to common chemotherapeutics can arise from several factors, including cell–cell adhesion and cell–matrix interactions. To assess the potential for reversal of 3D-induced chemoresistance, we tested ZJ-101 in combination with etoposide in MDA-MB-231 spheroids. Using a modified spheroid disassembly assay incorporating the cellular dyes calcein AM, Hoechst 33342, and ethidium homodimer (EthD), we are able to evaluate the effect of ZJ-101 on cell adhesion and proliferation. Similar to the GR assay, EthD allows for the determination of cytotoxicity. Using an 8 × 8 matrix, 63 combinations of ZJ-101 and etoposide are assessed (Figure 6A). Etoposide was found to be synergistic with ZJ-101 for the induction of cell death, as stained by ethidium homodimer (Figure 6A,B). Synergy was assessed using the SynergyFinder web app (https://synergyfinder.fimm.fi/ (accessed on 19 November 2020)) which calculates a ZIP synergy score for 8 × 8 combination matrices [16]. Etoposide synergizes with ZJ-101 with an average δ-score for concentrations above 10 μM between 40 and 50, indicating 40–50% excess synergy with ZJ-101 over etoposide alone (Figure 6D).

## 3. Discussion

Our group has identified several key phenotypes that suggest a distinct mechanism of action for the compound ZJ-101 by acting through the endomembrane system. We first identified the clear ability of ZJ-101 to cause cell rounding and dislodging from a growth substrate within a few hours. Cellular adhesion occurs in primarily two modalities: cell–cell and cell–substrate adhesion. To determine if ZJ-101 had preferential activity for either of these, we assessed its activity in a 3D spheroid model. To our amazement, extremely low doses of ZJ-101 could halt the formation of spheroids. At approximately 1 nM of the compound, we observed a halt in the progression of standard spheroid formation in the triple-negative breast cancer cell line MDA-MB-231. The disparity in EC_50s_ for the antiproliferative effect previously established versus anti-adhesion in the spheroid model was approximately 30-fold. This unexpected enhancement in potency against the spheroid model challenged our previous assumptions about the molecule.

We utilized the growth rate normalized metric assay (GR assay) to minimize variation between experiments and dependence on cell growth rates. A major benefit of this assay is its ability to discern cytostatic versus cytotoxic effects by comparing cell viability to cellular proliferation. Since all cell types have their own intrinsic growth rate kinetics, we can accurately compare responses to drugs between cell types by normalizing them to this rate. Using this live/dead cell image-based assay, we were able to determine that the effect of ZJ-101 on cancer cells is entirely cytostatic. In GR metrics, cytotoxic compounds are identified by their ability to deplete cell populations below initial seeding densities, denoted by a GR value less than 0. At a GR value greater than 0 but less than 1, cellular proliferation is suppressed in a cytostatic manner. ZJ-101 suppresses cellular growth with a GR value of 0.5 at the highest concentration of 1 μM using the calcein-AM/ethidium homodimer dye staining assay. We obtained the same EC_50_ value (30–90 nM) for ZJ-101 using GR metrics as we previously determined using resazurin-based viability assays (Appendix A). These results confirmed that one of ZJ-101′s primary novel phenotypes is cytostatic suppression of cellular proliferation.

Cytostasis is often a temporary cellular state induced by exogenous signals or chemicals that suppress cell division. Senescence, a physiological state of non-dividing cells that can be induced by oxidative stress or DNA damage, has all the hallmarks of cellular cytostasis. A key characteristic of senescence is a sharp increase in G1-phase cell populations. Because of the lack of change in any population (G1, S, or G2) during prolonged treatment with ZJ-101, cellular senescence could potentially be ruled out as a mechanism for the cytostatic activity. At time points of up to 48 h, MDA-MB-231 cells remained dislodged from their substrate under the sustained treatment of ZJ-101, further underscoring the distinct lack of any sub-G1 populations. Typically, cells will undergo apoptosis following sustained senescence or cell adhesion loss, a unique form of apoptosis termed anoikis [17]. To further explore senescence using GR metrics, we tested isogenic cell lines of HCT-116 containing knockouts of p21 and p53, key senescence potentiators [18]. We found no reversal of anti-proliferative activity under treatment with ZJ-101 in these knockout lines. Instead, p21 knockout potentiates the effect of ZJ-101′s suppression of cell proliferation. This result runs counter to an activation of senescence by ZJ-101, given that p21 expression is protective against cytostasis. Although the outcome of ZJ-101 treatment results in the halting of cellular proliferation, more work is required to determine the specific pathway(s) leading to this long-term cytostatic suppression.

To identify cellular pathways regulated by ZJ-101, transcriptome sequencing was undertaken across four doses in two contexts: before 2D adhesion loss at 6 h, and after adhesion loss at 24 h in 3D spheroids (Appendix A). Gene Ontology (GO) analysis identified increasing significance for the endoplasmic reticulum (GO:005783), cell adhesion (GO:007155), and the endomembrane system (GO:0012505) as the dose of ZJ-101 increases. Notably, endomembrane system genes represent the highest differentially expressed genes by significance between 2D and 3D culture formats. Genes related to the endoplasmic reticulum follow a paradoxical response to ZJ-101 treatment. Despite a general upregulation of heat-shock genes such as HSP90AA1, unfolded protein response (UPR) target genes were suppressed at 6 h. UPR-related genes remained suppressed at the 24 h time point as well, indicating a context-independent downregulation of ER-stress-responsive genes. Likewise, cell adhesion and extracellular matrix (ECM) genes were suppressed. Among these, several integrins, such as ITGA5, ITGAV, and ITGB5, were downregulated. These integrins are primarily responsible for cell–ECM adhesion and cannot solely explain the dual inhibition of cell–substrate and cell–cell adhesion caused by ZJ-101. Due to this unique transcriptional response, we reasoned that the stress caused by ZJ-101 may localize to a source within the wider endomembrane system, such as the Golgi and endosomes.

Genes involved in endosome homeostasis, such as ATP6V0A1, were significantly dysregulated by ZJ-101. V-type ATPases, such as ATP6V0A1, regulate endosome and vesicle pH. We compared ZJ-101 to the well-known inhibitor of V-type ATPases, Bafilomycin A1, through lysotracker staining, which fluoresces in the acidic compartments of endosomes and lysosomes. As observed in Figure 5B, bafilomycin A1 potently decreases lysotracker staining intensity relative to control, while ZJ-101 does not. Additionally, ZJ-101 treatment decreases endolysosome size in a similar manner to brefeldin A and kifunensine. Brefeldin A’s unique mechanism of redistributing the Golgi membrane to the ER through uncompetitive inhibition of the Arf1 GDP exchange cycle poses the most curious parallel for ZJ-101′s potential mechanism of action [19]. Although ZJ-101 does not redistribute GM130 as BFA does, they both likely enhance endolysosomal fission events, leading to the decreased puncta size observed after treatment. It is possible that ZJ-101 inhibits a target in opposition to the BFA mechanism, preserving the Golgi structurally but resulting in similar outcomes. Kifunensine also affects endolysosomal size, though it acts through inhibition of the ER mannosidase I enzyme. By inhibiting N-glycan trimming in the ER, protein sorting through the ER-associated degradation (ERAD) pathway is increased, leading to a loss of successful traversal of cargo through the secretory pathway [20].

We initially identified a dysregulation of glycosylation localized to the Golgi apparatus through lectin staining assays. Lectin staining determined a selectivity for decreasing N-acetyl-galactosamine (GalNAc) residues, which are typically added to substrates by GalNAc transferases within the Golgi [21]. Despite no clear structural alteration to the Golgi itself, GalNAc accumulation was suppressed at short time points by ZJ-101. Glycomics analysis confirmed a general downregulation of *O*- and *N*-glycan-bearing proteins (Appendix A), with several exceptions. Poly-sialylated high-molecular-weight N-glycans, bearing N-acetyl-lactosamine (LacNAc), were upregulated to a relatively high degree. These glycans are produced by the enzyme βGALT4 in the *trans*-Golgi, which indicates this region remains active during ZJ-101 treatment [22]. This result is counter to that of the transcriptomic signature, where transcripts of B4GALT1 and other galactosyltransferases are found to be suppressed by ZJ-101. It is possible this downregulation is caused by a feedback loop from these LacNAc residues accumulating in the *trans*-Golgi. Additional poly-sialylated O-glycan residues, such as the sialyl-Tn antigen, known to be processed by ST6GalNAc, were also found to be upregulated (Table 1), indicating that sialyl-transferase activity in the Golgi also remained. Despite the global decrease in N-glycan abundance, lectin staining of both high-mannose and N-acetyl-glucosamine (GlcNAc) residues was unchanged by ZJ-101 treatment. This discrepancy may stem from the long half-lives of proteins bearing these residues, which may not yet have turned over. It’s likely that new proteins exiting the Golgi, or held within the trans-Golgi network, bear the more complex LacNAc residues identified in the glycomics analysis. A similar accrual of poly-sialylated N-glycans in the trans-Golgi was recently observed by Kitano et al. following Rab11 knockdown, which may provide a possible explanation for ZJ-101′s effect [23]. Further inquiry is required to determine whether glycosyltransferase localization at the Golgi is affected by ZJ-101.

We performed a high-content 3D-spheroid drug combination synergy assay using a modified spheroid disassembly assay. Our objective was to identify whether the anti-adhesive effect of ZJ-101 could reverse a common mechanism of drug resistance mediated by tight cellular adhesion [24,25]. In a 2D context, etoposide treatment induces high BRCA1 expression in MDA-MB-231 cells, making them naturally resistant to etoposide’s mechanism of action [26]. For MDA-MB-231 spheroids in particular, ECM interactions further limit drug accessibility, providing a key context to assess 3D-induced chemoresistance to topoisomerase II inhibitors [27]. Broad suppression of extracellular matrix and cell adhesion gene transcripts by ZJ-101 lends further support to a possible reversal of chemoresistance mediated by such factors. Our combination synergy assay utilizes the zero interaction potency (ZIP) synergy score, which evaluates individual drug combination pairs relative to their separate dose–response curves [28]. ZIP synergy normalizes drug response to assume a “zero interaction” or minimal shift in dose–response between combination pairs. Etoposide synergizes with ZJ-101 with an average δ score of 10.5 across all combination pairs, indicating positive synergy. δ-scores describe the excess percentage at which combination pairs are synergistic or antagonistic. For etoposide, the average δ-score for concentrations above 10 μM is between 40 and 50, indicating 40–50% excess synergy with ZJ-101 over etoposide alone.

In conclusion, we have identified several unique phenotypes produced by treatment with the marine natural product-derived compound, ZJ-101. We discovered a strongly cytostatic and antiadhesive phenotype at single-digit nanomolar concentrations, which is unique and unprecedented. Through transcriptomic analysis, we identified dysregulation of the endomembrane system, which was later confirmed by lectin staining and glycomics analysis. Our work has established a potential synergistic mechanism for the reversal of chemoresistance mediated by 3D cell adhesion. Other mechanisms of multidrug resistance mediated by endolysosomal trafficking provide a future direction for compounds such as ZJ-101, which dysregulate glycosylation [29]. Recent screens for regulators of cellular glycosylation have indicated potential use against SARS-CoV-2 viral entry, offering another potential path for further ZJ-101 investigation [30].

## 4. Materials and Methods

### 4.1. Cell Culture

Cells were cultured in DMEM media (Gibco Cat # 11885, Billings, MT, USA) supplemented with 10% fetal bovine serum and 1% penicillin/streptomycin.

### 4.2. Cell Cycle Analysis

Cell cycle analysis was performed using propidium iodide staining of fixed MDA-MB-231 cells treated with ZJ-101. Briefly, a 70% confluent 10 cm dish of MDA-MB-231 was collected via trypsinization and washed with 1× PBS. Pelleted cells were fixed by the dropwise addition of 2 mL of ice-cold 75% EtOH. Fixed cells were washed again with PBS and stained with 1 mg/mL propidium iodide solution prior to FACS analysis.

### 4.3. Growth-Rate Inhibition Metric Analysis

Control populations of MDA-MB-231 cells were prepared for growth rate normalization at seeding densities of 50, 200, 500, and 1000 cells per well in a 384-well flat bottom plate (Corning Inc., Corning, NY, USA). Following compound treatment, cells were stained with 1 μM Calcein AM (for live cells), 20 μg/mL Hoechst 33342 (for nucleus), and 1 μM ethidium homodimer (a cell membrane impermeable dye for dead cells) for 15 min at 37 °C prior to imaging. Imaging was performed using the 4× objective on the ImageXpress Micro (Molecular Devices, San Jose, CA, USA) with image-based focusing. Collected images were assessed using the live/dead program in MetaXpress (version 6.1), and data were organized and entered into grcalculator.org, where growth rate normalized inhibition calculations were performed for each compound tested.

### 4.4. 3D Spheroid Assays

Three-dimensional spheroid formation was assessed using Corning ultra-low attachment (ULA) plates. MDA-MB-231 cells were seeded in ULA plates at a density of 2 × 10^3^ cells/well (96-well) or 5 × 10^2^ cells/well (384-well) in DMEM media supplemented with 10% FBS and centrifuged for 5 min at 400× *g*. Cells were then left undisturbed in a 37 °C, 5% CO_2_ incubator for 72 h to form spheroids. ZJ-101 was added at the indicated concentrations and incubated for a further 72 h. Pretreatment of ZJ-101 was also performed with basic light images taken every 24 h for up to 72 h. Spheroidicity was determined based on the overall diameter normalized to an untreated control using ImageJ (version 1.52r).

The 3D spheroid combination assay was performed as a spheroid disassembly assay requiring 72 h of pre-formed spheroids prior to a further 72 h of drug addition. ZJ-101 and etoposide were arranged with 1:3 dilutions in an 8 × 8 matrix format at the indicated concentrations. Fluorescent images were acquired after staining using the same protocol and instrumentation outlined in the GR assay. Laser-based focusing was utilized to obtain clear spheroid images from the bottom of each well. Ten z-stack images were taken and combined into a single maximum-intensity image. Images were processed with the live/dead program in MetaExpress.

### 4.5. Cell Staining and Imaging

Basic light microscopy was performed on HEK293T cells cultured in 6-well TC-treated plates (Greiner #657160, Kremsmünster, Austria) by capturing images through the 20X objective of a Zeiss Axiovert 25 (Carl Zeiss Microscopy, White Plains NY, USA) with a 12.2 megapixel camera. For MDA-MB-231 spheroid light microscopy, the same protocol was used with a 10X objective.

For all staining assays, HeLa cells were cultured on 96-well CellView TC-treated microplates (Greiner #655891, Kremsmünster, Austria). Cells were washed with cold PBS prior to fixation with 4% PFA for 10 min at RT. After fixation, cells were washed twice with PBS and permeabilized with 0.1% Triton-X for 10 min at RT. Cells were again washed twice with PBS and blocked with cell staining buffer for 30 min. Antibodies for GM130 (Cell Signaling #12480, Danvers, MA, USA) were added at 1:1000 in staining buffer for 1 h at RT. After three washes with PBS, secondary anti-rabbit AlexaFluor-488, or AlexaFluor-647 (ThermoFisher, Waltham, MA, USA) were added along with the indicated fluorescent lectin conjugates (HPA-647, WGA-488, and PNA-555 from ThermoFisher) at 1:1000 dilution for 1 h RT. Cells were again washed and stained with 1:10^4^ Hoechst 33,342 for 3 min prior to imaging with an OperaPhenix (PerkinElmer, Waltham, MA, USA). Images were uploaded to the Columbus Analyzer (version 2.9.1.699) and processed for high-content analysis.

For the live cell lysotracker assay, HeLa cells were cultured as above during treatment with the specified compounds. Cells were loaded with DMEM media containing 100 nM Lysotracker Deep Red for 30 min at 37 °C. After lysotracker staining, media was exchanged for Live Cell Imaging Solution (Invitrogen #A14291DJ, Waltham, MA, USA) containing 1:10^4^ Hoechst 33342 for nuclei staining. Images were uploaded to the Columbus Analyzer and processed for high-content analysis using default settings for spot detection and intensity calculation.

### 4.6. Transcriptome Analysis

MDA-MB-231 cells from 10 cm dishes were harvested through scraping (for 2D) or spheroids collected (for 3D) using a wide-gauge pipette and subjected to RNA extraction via the RNeasy mini kit using the manufacturer’s instructions. Biological replicates of N = 3 were used for both sets of analyses, with N = 96 spheroids representing a single biological replicate for the 3D populations. RNA-sequencing was performed by Genewiz as paired-end 150 bp reads following poly-A selection to enrich mRNA transcripts. Paired-end FASTQ files were uploaded to Galaxy using the public server at usegalaxy.org (accessed on 7 April 2021) and aligned to hg38 using HISAT2. Transcripts were assembled and counted using htseq-count, and differential gene expression was evaluated with DESeq2 with default settings. Fold changes were assessed against the DMSO vehicle controls. Transcripts were annotated using the most current GENCODE release. Gene Ontology analysis was performed using g:Profiler at https://biit.cs.ut.ee/gprofiler/gost (accessed on 8 April 2021) by inputting the top 200 significant genes for each concentration of ZJ-101 and arranging them by descending order of significance [9]. Heatmaps and PCA plots were generated using ClustVis v1.0 at https://biit.cs.ut.ee/clustvis/ (accessed on 8 April 2021) [10].

The data discussed in this publication have been deposited in NCBI’s Gene Expression Omnibus and are accessible through GEO accession number GSE231359 https://www.ncbi.nlm.nih.gov/geo/query/acc.cgi?acc=GSE231359 (accessed on 8 April 2021) [31].

### 4.7. Glycan Analysis

#### 4.7.1. Glycan Incorporation Assay

Briefly, 50 μM azido-modified sugars (Invitrogen) tetraacetylated N-azidoacetylglucosamine (GlcNAz), tetraacetylated N-azidoacetylgalactosamine (GalNAz), tetraacetylated N-azidoacetyl-D-mannosamine (ManNAz), and 100 μM alkynyl-fucose (Invitrogen) were added to 96-well plates in combination with the indicated doses of ZJ-101. Cells were then washed, fixed, and permeabilized prior to the click reaction. Copper-catalyzed click reactions were performed using the Click-iT Cell Reaction Buffer Kit (Invitrogen C10269) per the manufacturer’s instructions, containing 5 μM of either Fluor alkyne-647 (Invitrogen A10278) to label azido-incorporated sugars or Fluor Azide-488 to label incorporated alkynyl-fucose. Plates were washed five times before Hoechst counterstaining and imaging. Glycan incorporation was determined by the total intensity of each signal normalized to untreated control cells.

#### 4.7.2. Glycome Profiling

Glycomics profiling was performed by Creative Proteomics. N-glycans were prepared from fresh cell pellets washed with PBS, resuspended in 1 mL of lysis buffer, and sonicated (5 pulses of 10 s). Samples were then dialyzed in 50 mM ammonium bicarbonate for 24 h at 4 °C, with the buffer changed three times. Following dialysis, the samples were lyophilized. To the lyophilized powder, 1 mL of 2 mg/mL DTT was added, and the solution was incubated at 50 °C for 2 h. Briefly, 1 mL of a 12 mg/mL IAA (iodoacetamide, Sigma, St. Louis, MO, USA) solution was then added and incubated at RT in the dark for 2 h. The DTT and IAA-treated proteins were then dialyzed against 50 mM ammonium bicarbonate (Sigma) for 24 h at 4 °C. Samples were next resuspended in 1 mL of 500 µg/mL TPCK-treated trypsin (Sigma) solution and incubated at 37 °C overnight. The trypsin reaction mixture was purified over C18 Sep-Pak columns (Waters, Milford, MA, USA) by 1-propanol elution. Fractions containing peptides were pooled and lyophilized. The lyophilized peptides were resuspended in 200 µL of 50 mM ammonium bicarbonate, to which 3 µL of PNGaseF (New England Biolabs, Ipswich, MA, USA) was added for a 4 h incubation at 37 °C. Following this initial incubation, another 5 µL of PNGaseF was added for overnight incubation at 37 °C. The enzymatic reaction was stopped by the addition of two drops of 5% acetic acid, and the released N-glycans were purified over C18 Sep-Pak columns. Flow-through and 5% acetic acid washing fractions containing the released N-glycans were pooled and lyophilized and were subject to permethylation. For O-glycan analysis, the PNGaseF-treated glycopeptides were eluted from the C18 column with 1-propanol. The lyophilized eluted peptides were subjected to *O*-glycan preparation.

*O*-glycan-containing powder was solubilized by 400 µL of a sodium borohydride (Sigma-Aldrich) solution in 0.1 M NaOH (55 mg NaBH4/1 mL 0.1 M NaOH) and incubated overnight at 45 °C. The reaction was stopped by adding drops of pure acetic acid until the fizzing stopped. The samples were passed through a Dowex 50W X8 resin (Sigma-Aldrich) column, and the pooled fractions were dried by lyophilization. The lyophilized samples were next resuspended in 1 mL of an acetic acid:methanol solution (1:9 *v*/*v*) and co-evaporated under nitrogen flow. This step was repeated 3 more times, and the dried samples were resuspended in 200 μL of 50% methanol prior to being loaded onto the C18 Sep-Pak column. Free *O*-glycans were collected in the flow-through and 5% acetic acid wash fractions. These fractions were pooled, lyophilized, and subjected to permethylation.

Permethylation was performed as follows. Seven pellets of NaOH in 3 mL of DMSO were ground with a mortar and pestle. One milliliter of this slurry solution was added to the dry sample in a glass tube with a screw cap. Five hundred microliters of iodomethane was then added to the slurry and shaken at RT for ~30 min. After the reaction reaches completion, noting the formation of white solids, the cap is released slowly to relieve the gas pressure that has built up. One milliliter of MilliQ water was added to stop the reaction, and the sample was vortexed until all solids were dissolved. To the sample, 1 mL of chloroform and an additional 3 mL of MilliQ water were added with continuous vortexing to mix both phases. The samples were then centrifuged briefly to separate the chloroform and the water phases (~5000 rpm, <20 s). The aqueous top layer was discarded, and washing was repeated twice with the addition of 3 mL of MilliQ water. The chloroform fraction was then dried with a SpeedVac (~20–30 min). A C18 Spe-Pak (200 mg) column was prepared with methanol, MiliQ water, acetonitrile, and MilliQ water. The dry sample was resuspended with 200 µL of 50% methanol and loaded onto the column. The column is then washed with 2 mL of 15% acetonitrile and eluted into a clean glass tube with 3 mL of 50% acetonitrile. Finally, the eluted fraction was subjected to MS analysis.

MS data were acquired on a Bruker UltraFlex II MALDI-TOF mass spectrometer instrument (Bruker Scientific LLC, Billerica, MA, USA). Reflective positive mode was used, and data were usually recorded between 500 *m*/*z* and 6000 *m*/*z* for *N*-linked glycans and between 500 *m*/*z* and 4000 *m*/*z* for *O*-glycans. For each MS N- and *O*-glycan profile, the aggregation of 20,000 laser shots or more was considered for data extraction. Mass signals with a signal/noise ratio of at least 2 were considered, and only MS signals matching an *N*- and *O*-glycan composition were considered for further analysis and annotation. Subsequent MS post-data acquisition analysis was made using mMass [32].

### 4.8. Statistical Analysis

Statistical tests were performed on GraphPad Prism version 9.5. The normalized data from *n* = 3 independent experiments were analyzed for significance using the Brown–Forsythe and Welch one-way ANOVA. Statistical significance is set for * *p*  <  0.05, ** *p*  ≤  0.01, *** *p*  ≤  0.001.

## Figures and Tables

**Figure 1 ijms-24-09575-f001:**
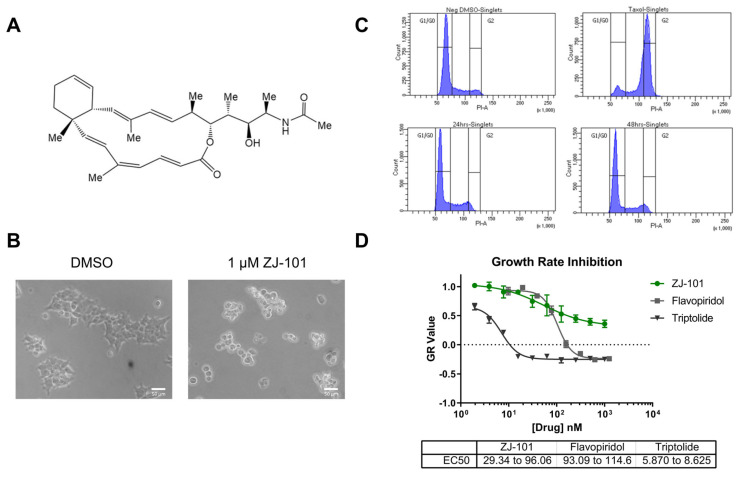
ZJ-101 is cytostatic and anti-adhesive. (**A**) Structure of ZJ-101. (**B**) Cell morphology after treatment with ZJ-101. (**C**) Cell cycle analysis via propidium iodide (PI) staining in MDA-MB-231 cells. Taxol utilized as a G2-phase block control. (**D**) Growth rate inhibition metrics assay (GR assay) determines that ZJ-101 is primarily cytostatic with an EC_50_ between 29 and 96 nM. when compared to known cytotoxic compounds.

**Figure 2 ijms-24-09575-f002:**
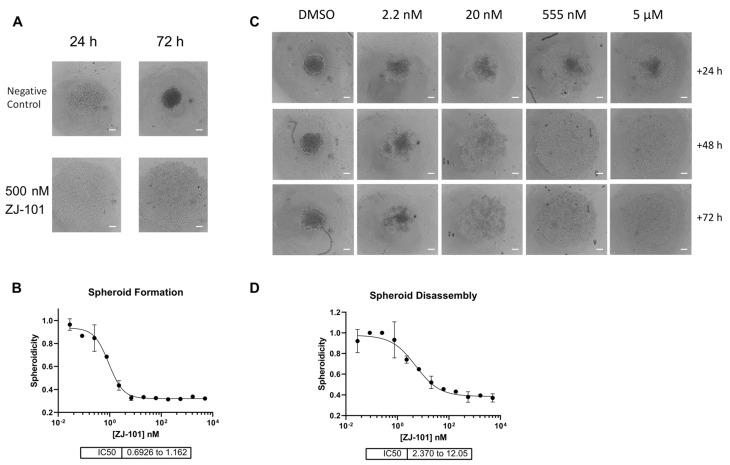
Three-dimensional cell adhesion is potently disrupted by ZJ-101. (**A**) MDA-MB-231 spheroid formation assay. Scale bar = 200 μm (**B**) Effect on spheroid formation is dose-dependent, with an IC_50_ of approximately 1 nM. (**C**) Representative images of pre-formed spheroids (72 h old), treated with ZJ-101 over the indicated times. (**D**) Effect on spheroid disassembly has an EC_50_ of approximately 5 nM.

**Figure 3 ijms-24-09575-f003:**
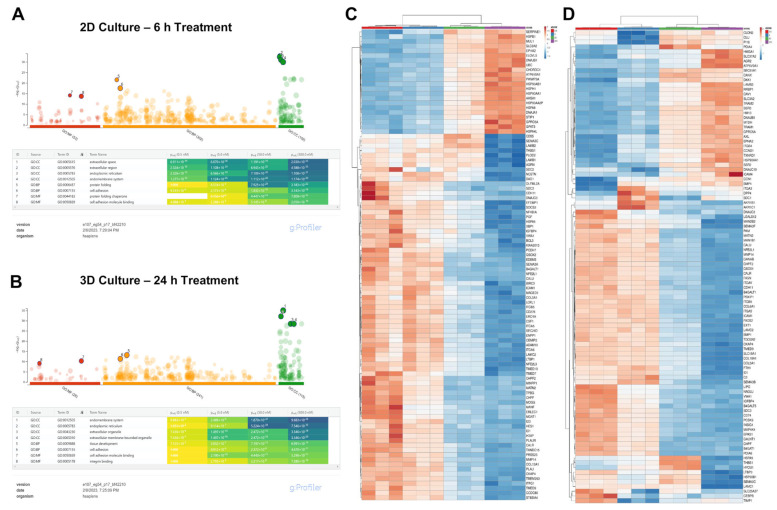
Endomembrane system genes differentially regulated independent of cellular context. (**A**) g:Profiler Gene Ontology (GO) term enrichment analysis after 6 h treatment of four ZJ-101 doses (0.5, 5.0, 50, 500 nM) in MDA-MB-231 cells cultured in 2D format (*n* = 3 biological replicates). The top significantly enriched GO terms from each category were selected and include the extracellular space, endoplasmic reticulum, and endomembrane system with increasing dose. Terms which do not reach significance at a particular dose are indicted by strikethrough (**B**) GO terms for 3D-cultured MDA-MB-231 cells (*n* = 3 biological replicates) treated with ZJ-101 for 24 h. (**C**) ClustVis heatmap for top differentially expressed genes after 6 h ZJ-101 dose–response in 2D-cultured MDA-MB-231 cells. (**D**) ClustVis heatmap for the 24 h ZJ-101 dose–response in 3D-cultured MDA-MB-231 cells.

**Figure 4 ijms-24-09575-f004:**
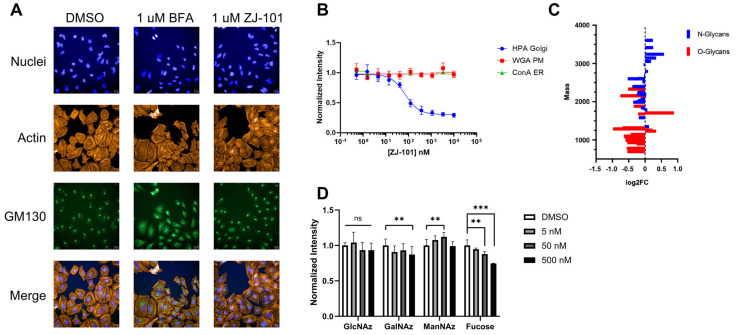
Golgi function is dysregulated by ZJ-101. (**A**) Golgi structure within HeLa cells, as visualized by GM130 immunofluorescence, is not altered by ZJ-101 treatment. Brefeldin A (BFA) has a dose-dependent effect on GM130 redistribution to the endoplasmic reticulum (ER) (*n* = 3). Scale bar: 20 µM. (**B**) Lectin staining imaging assay; helix pomatia agglutinin (HPA), wheat germ agglutinin (WGA), and concanavalin A (ConA). ZJ-101 selectively and dose-dependently decreases HPA staining intensity at 4 h of treatment. (*n* = 3). (**C**) Glycomics analysis sorted by mass for N- and O-glycans after 6 h treatment with 500 nM ZJ-101. Most N- and O-glycans are downregulated, notable high-molecular weight N-glycans and complex O-glycans are upregulated and noted in Table 1. (**D**) Glycan incorporation imaging assay performed using biorthogonal click chemistry of azide-modified glucosamine, galactosamine, mannosamine, and alkynyl-modified fucose. ZJ-101 treatment of 6 h results in significant decreases in alkynyl-Fuc and azido-GalNAc incorporation (*n* = 3). Data represent the mean ± SEM. n.s. = not significant, ** *p* ≤ 0.01, *** *p* ≤ 0.001.

**Figure 5 ijms-24-09575-f005:**
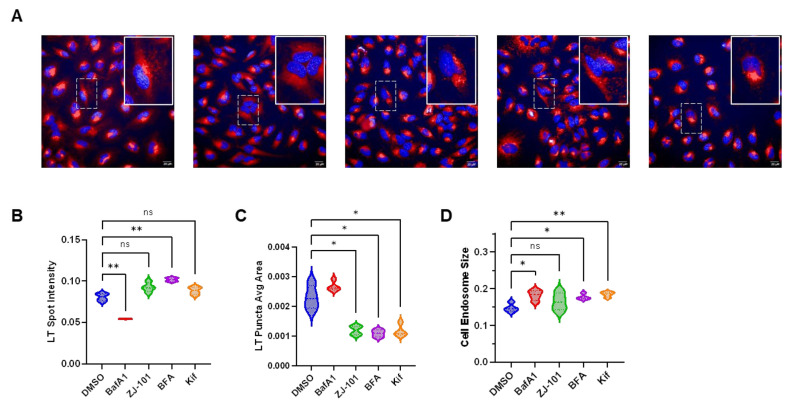
Endolysosomes are modulated by ZJ-101. (**A**) Representative images of lysotracker staining with LT dye in red, nuclei in blue. Cells were treated for 2 h with the following doses: 200 nM bafilomycin A1, 500 nM ZJ-101, 500 nM brefeldin A, 10 μM kifunensine. (**B**) LT spot intensity for each compound assessed. Bafilomycin A1 significantly decreases intensity relative to vehicle control. (**C**) LT puncta average size by area. Treatment with inhibitors of various endomembrane system components results in smaller than average LT puncta spot size, while cells treated with bafilomycin A1 do not have significantly changed LT puncta area relative to DMSO. (**D**) Total LT stained endolysosomal area within the cell. Small, but significant, increases in the total size of endolysosomal compartments are seen with brefeldin A, kifunensine, and bafilomycin A1. ZJ-101 treatment does not result in significant changes to overall endolysosomal area. n.s. =  not significant, * *p* < 0.05, ** *p* ≤ 0.01.

**Figure 6 ijms-24-09575-f006:**
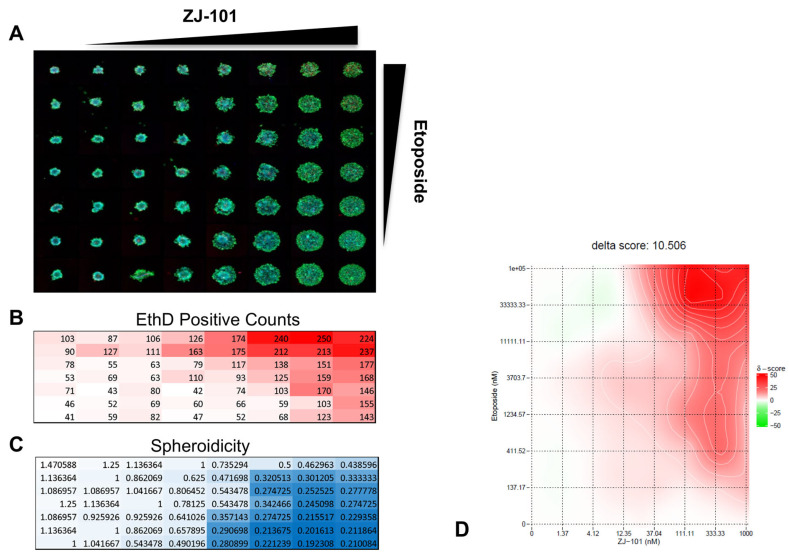
ZJ-101 reverses resistance to etoposide in combination with matrix spheroid assay. (**A**) Fluorescent live-cell images of MDA-MB-231 3D spheroid matrix of ZJ-101 and etoposide at 1:3 dilutions from their maximum concentrations of 1 μM and 100 μM respectively. (**B**) Ethidium homodimer (EthD) counts for each well shown. White–Red heatmap corresponds with increasing counts (**C**) Spheroidicity—the measurement of the diameter of each spheroid normalized to untreated wells. Lower spheroidicity indicates disruption of cell–cell adhesion caused by ZJ-101 treatment and is indicated by the White–Blue heatmap. (**D**) SynergyFinder graph of combinations of ZJ-101 and etoposide shows enhanced synergy above 10 μM etoposide. Green-Red heatmap of delta-scores indicate negative to positive regions of synergy respectively for all dose combination pairs.

**Table 1 ijms-24-09575-t001:** Upregulated glycans.

*N*-Linked Glycans			
Composition	Proposed Structure	Observed *m*/*z*	log2FC
Hex:6 HexNAc:5 NeuAc:2	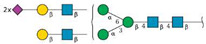	3241.6	0.579
Hex:7 HexNAc:6 Fuc:1	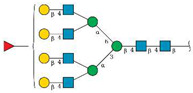	3142.6	0.341
Hex:6 HexNAc:5 Fuc:1 NeuAc:2	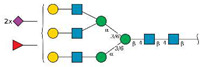	3415.7	0.249
Hex:6 HexNAc:5 NeuAc:3	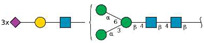	3602.8	0.241
Hex:6 HexNAc:5 Fuc:1 NeuAc:1	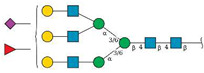	3054.5	0.183
***O*-Linked Glycans**			
**Composition**	**Proposed Structure**	**Observed *m*/*z***	**log2FC**
Hex:2 HexNAc:2 NeuAc:2	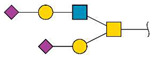	1705.9	0.883
Hex:2 HexNAc:3	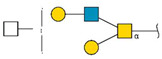	1228.6	0.338
Hex:1 HexNAc:1 NeuAc:2	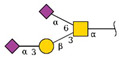	1256.6	0.229

## Data Availability

Data are available upon request.

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
