# Peer review of "Modulation of the Endomembrane System by the Anticancer Natural Product Superstolide/ZJ-101"

_ijms, 2023, doi:10.3390/ijms24119575_

Round 1

Reviewer 1 Report

Comments to Authors

The manuscript entitled, “Modulation of the Endomembrane System by the Anticancer Natural Product Superstolide/ZJ-101” by Sanchez et al., shows the effect of ZJ-101 on cell growth, cell adhesion, up-regulation of genes of the endosome system, glycan structures, dysregulation of Golgi apparatus, and drug resistance. Although the manuscript is of great of interest, the data need to be strengthened to support their proposed mechanism of action for ZJ-101. The paper should address the following points:

The descriptions of the various experiment in the results are inadequate in the text of results section, and furthermore results should not be included in figure legends. Further figure legends are poor to fair since appropriate detail is lacking. Cell proliferation assays should be conducted, and spheroids should be made using 5D 3-D plates. The concentrations of the ZJ-101 are inconsistent between text of the results and figure legends. It is unclear which cell lines were used for the various experiments, particularly from section 2.4 and onward. In terms of Fig1B it would be more appropriate to perform this with MDA-MB-231 since authors use this cell line for the transcriptome analysis. The various drugs and lectins need to be defined and references provided at the start of using them for the respective experiment. For instance, WGA does bind O-glycans, not just N-glycans. Also, ConA binds high mannose N-glycans in membranes other than the Golgi apparatus, such as plasma membrane. Also clearly state whether using live or dead cells in results. Improve discussion by elaborating on the results found within, not just restate results. Materials and methods lack full description of experimental acquisition and analysis. For instance, how is the combination matrix spheroid assay conducted. The quality of the figures is poor, including Figs 1C, 2A, 2C, Fig 3, Fig 6D and the remainder of panels in fig 6 is fair.

Other points:

5D 3-d cell culture dishes, instead of ULA plates, should be used for spheroid studies since they make uniform spheroids. Images of spheroids are unclear. Are spheroids breaking apart or just going from a tight to loose spheroid as dispersed cells are not visible. Panel C need initial spheroid used. Also state whether the image at day +1, +2 and +3 are the same spheroid. Data from fig 2C should be summarized in a graph. Panel 2B show how you get the measurement as it may be better to measure area since diameter is inconsistent. Cell growth results could be strengthened by measuring cell proliferation. Fig 3 is of poor quality and not described well in the results. The function of the five genes up-regulated should be mentioned as it is not clear from reading text of results that they are part of the endomembrane system. Brefeldin should be defined here, not later. The rationale for 2D -6 h treatment is unclear to the reader since fig1 used HEK293 cells. Also the choice of 24 h for spheroid is unclear since day 0 is not shown. Figure legend of figure 4 lacks explanation. What cell line was used? Also, explain measurement better. Figure 5 intensity measurement should be described better as experiment is unclear. What cell line was used? Results section for figure 6 is inadequate as description is too brief.

Reviewer 2 Report

This article presented marine natural product who have anticancer activities. The authors has identified several key phenotypes which suggest a distinct mechanism-of-action for the compound ZJ-101 by acting through the endomembrane system. This study is very interesting and useful.

 Dear Author!

 Please revisit the article with reg. no.2414433  in accordance with Editor’s comments:

·         I recommend that the authors split Figure 3 into two or three separate figures. Poor quality of Figure 3!

·         General conclusions are missed. Must be added!

Reviewer 3 Report

I have evaluated the manuscript (IJMS-2414433) titled “Modulation of the Endomembrane System by the Anticancer Natural Product Superstolide/ZJ-101 by Liu and coworkers. I found this article interesting for the readers and followed the journal IJMS’ scope. I don’t have any major comments, however, the author needs to focus on presentation of the research work with data and proper discussion to make this article more interesting to the reader of IJMS. The supplemental document should not be in power point.

I would recommend the article be published in IJMS after minor corrections. 

The author needs to address the following comments/corrections.

1.    Table of content of SI is missing, and title of the manuscript with authors.

2.    All the figures and tables in the SI need titles and numbers with footnotes.

3.    The author should correct the format of references wherever needed (e.g Year Bold, Volume Italic etc).

4.    The author should include the full form of abbreviation when first used in the text.

5.    The author needs to clear figure 1, it is a bit messy.

6.    The resolution of all figures needs to be improved.

Reviewer 4 Report

This is a well-crafted paper that will attract interest amongst the natural product and pharmaceutical development communities.  The results are clearly described, and reveal a novel mechanism of action for this natural product analogue, something that may have considerable potential.  As a chemist, I can mainly comment on the chemical content, which is sound.  The details of the biological experiments and critical analysis of results are beyond my expertise.

One minor point: chemical compounds are not normally capitalized, unless of course at the start of a sentence.

Overall, this is interesting work that merits publication in its present form.

Round 2

Reviewer 1 Report

Good Job.